# Design and Fabrication of a Film Bulk Acoustic Wave Filter for 3.0 GHz–3.2 GHz S-Band

**DOI:** 10.3390/s24092939

**Published:** 2024-05-05

**Authors:** Chao Gao, Yupeng Zheng, Haiyang Li, Yuqi Ren, Xiyu Gu, Xiaoming Huang, Yaxin Wang, Yuanhang Qu, Yan Liu, Yao Cai, Chengliang Sun

**Affiliations:** 1The Institute of Technological Sciences, Hubei Key Laboratory of Electronic Manufacturing and Packaging Integration, Wuhan University, Wuhan 430072, China; gaochao96@whu.edu.cn (C.G.); 2019300003014@whu.edu.cn (Y.Z.); lhy2022@whu.edu.cn (H.L.); renyuqi@whu.edu.cn (Y.R.); guxiyu@whu.edu.cn (X.G.); hxmmmm@whu.edu.cn (X.H.); wang93@whu.edu.cn (Y.W.); quyuanhang@whu.edu.cn (Y.Q.); liuyan92@whu.edu.cn (Y.L.); 2Hubei Yangtze Memory Laboratories, Wuhan 430205, China; 3Wuhan Institute of Quantum Technology, Wuhan 430072, China

**Keywords:** film bulk acoustic-wave resonator, film bulk acoustic-wave filter, acoustic–electromagnetic simulation, power capacity

## Abstract

Film bulk acoustic-wave resonators (FBARs) are widely utilized in the field of radio frequency (RF) filters due to their excellent performance, such as high operation frequency and high quality. In this paper, we present the design, fabrication, and characterization of an FBAR filter for the 3.0 GHz–3.2 GHz S-band. Using a scandium-doped aluminum nitride (Sc_0.2_Al_0.8_N) film, the filter is designed through a combined acoustic–electromagnetic simulation method, and the FBAR and filter are fabricated using an eight-step lithographic process. The measured FBAR presents an effective electromechanical coupling coefficient ($k_{eff}^{2}$) value up to 13.3%, and the measured filter demonstrates a −3 dB bandwidth of 115 MHz (from 3.013 GHz to 3.128 GHz), a low insertion loss of −2.4 dB, and good out-of-band rejection of −30 dB. The measured 1 dB compression point of the fabricated filter is 30.5 dBm, and the first series resonator burns out first as the input power increases. This work paves the way for research on high-power RF filters in mobile communication.

## 1. Introduction

The evolution of wireless communication continuously drives filters toward high-frequency and high-power applications [1,2,3,4]. Benefiting from the characteristics of high frequency, high quality, and small size, film bulk acoustic-wave resonator (FBAR) filters have captured more and more attention in the radio frequency (RF) filter market [5,6]. Aluminum nitride (AlN) has been commonly employed in the design and fabrication of FBAR due to its high longitudinal sound velocity (11,354 m/s) and low material loss [7]. However, the limited intrinsic electromechanical coupling coefficient affects the bandwidth of the filter, constraining further development of the filter [8]. Doping AlN with additional elements has been demonstrated to enhance its piezoelectric response [9,10,11,12,13,14], with examples including V [9,12], Ta [9], Ti [10], Mn [11], Sc [13,14], and others. Specifically, Sc_0.43_Al_0.57_N has been proven to increase the piezoelectric coefficient by five times that of AlN [13]. Meanwhile, research results suggest that even with a Sc concentration of 50 at.% in ScAlN, the longitudinal sound velocity of the ScAlN still exceeds that of ZnO (6080 m/s) [15]. Therefore, Scandium-doped aluminum nitride (ScAlN) is recognized for its exceptional properties, including a high electromechanical coupling coefficient and high sound velocity [13,16,17,18]. These attributes make ScAlN thin films highly suitable for filter fabrication, thus establishing them as a prominent focus in the realm of scientific research [7,19,20].

Recently, researchers have conducted extensive research on ScAlN-based FBARs. Ramin Matloub et al. presented a Sc_0.1_Al_0.9_N-based FBAR with a resonant frequency of around 2.5 GHz and an effective electromechanical coupling coefficient ($k_{eff}^{2}$) of 7.3% [21]. Milena Moreira et al. demonstrated a Sc_0.15_Al_0.85_N-based FBAR operating at 2.15 GHz with a $k_{eff}^{2}$ of 12.07% [17]. Mingyo Park et al. achieved multi-GHz FBARs for the first time on a single-crystalline ScAlN/Mo stack, generating a remarkably high *f* × *Q* value of 5.64 × 10^12^ Hz and an *f* × *Q* × $k_{t}^{2}$ value of 8.33 × 10^10^ Hz [22]. Jialin Wang et al. reported an FBAR based on a Sc_0.3_Al_0.7_N thin film, with a $k_{eff}^{2}$ of 18.1% and a quality factor (*Q*) of 210 [23]. Jialin et al. fabricated a polarization-switchable FBAR based on Sc_0.3_Al_0.7_N, with a fundamental resonance frequency of 3.17 GHz and a $k_{eff}^{2}$ of 11.4% [24]. Suhyun Nam et al. presented a mm wave trilayer AlN/Sc_0.3_Al_0.7_N/AlN higher-order-mode FBAR with a $k_{eff}^{2}$ of 5.2% in fundamental mode (GHz) and a $k_{eff}^{2}$ of 5.5% at higher-order response (31 GHz) [25]. Yang Zou et al. proposed a Sc_0.2_Al_0.8_N-based FBAR filter, for which the measured center frequency was 4.25 GHz, and the bandwidth of the filter was 189 MHz [26]. Wentong Dou et al. designed and fabricated an FBAR using a Sc_0.3_Al_0.7_N thin film, and the $k_{eff}^{2}$ was measured to be 17.8% at 4.75 GHz [27]. However, these studies are mainly focused on the high operating frequency and high effective electromechanical coupling coefficient of FBARs. There is considerable potential for advancements in exploring the power characteristics of FBARs, which is particularly crucial in 5G and 6G communications.

In this paper, a film bulk acoustic-wave filter is designed for the 3.0 GHz–3.2 GHz S-band [28], and the power characteristics of the filter are also investigated. The combined acoustic–electromagnetic method is adopted to achieve a more accurate simulation of the filter. Additionally, the temperature distribution during filter operation is investigated. Then, the proposed FBAR and filters are fabricated by an eight-mask layer process. Using high-quality Sc_0.2_Al_0.8_N film with a well-oriented *c*-axis, we achieve an FBAR with a $k_{eff}^{2}$ value up to 13.3%. The fabricated filter exhibits a −3 dB bandwidth of 115 MHz (from 3.013 GHz to 3.128 GHz), a low insertion loss of −2.4 dB, and a good out-of-band rejection of −30 dB. Furthermore, a power capacity testing system is set up, and the 1 dB compression point of the fabricated filter is 30.5 dBm. The first series resonator burns out first as the power increases, which matches the results obtained from the temperature distribution simulation. This study offers potential pathways for the development of high-power RF filters, thereby benefiting the research on filters for 5G and 6G communications.

## 2. Design and Fabrication

The filter design begins with circuit simulation using the Mason model [29], which is derived from the one-dimensional acoustic wave equation [30]. As shown in Figure 1a, the nonpiezoelectric layers are represented by a transmission line model. Meanwhile, an additional transformer is incorporated within the transmission line for the piezoelectric layer, serving to illustrate the mutual conversion between electrical and mechanical energy in the circuit. In this model, *Z* (*Z* = ρ·v) represents the acoustic impedance, *k* (*k* = 2πf/v) is the wave number, and *d* is the thickness of each layer. It is evident that the Mason model can effectively simulate the impact of layer thickness and material parameters, encompassing mass density (ρ), velocity (v), stiffness (*C*), etc. The key material parameters of Sc_0.2_Al_0.8_N used in the simulation can be found in [31].

As shown in Figure 1b, the designed FBAR consists of a Sc_0.2_Al_0.8_N piezoelectric film sandwiched between the top and bottom electrodes (Mo). A bandpass filter can be obtained by cascading series and parallel FBARs [30]; in this work, the four-stage ladder topology shown in Figure 1b is adopted. The frequency characteristics of both series and parallel resonators can be finely tuned by adjusting the thickness of each layer. In this study, the target frequency range of the filter is 3.0 GHz–3.2 GHz, with a center frequency of 3.1 GHz. As illustrated in Figure 1c, the series resonant frequency (*f_s_*) of the series resonator and the parallel resonant frequency (*f_p_*) of the parallel resonator are precisely tuned to align with the center frequency of the target filter band. Detailed thickness information for each layer is shown in Table 1.

Figure 1d,e demonstrate the simulated electrical response of the filter using the Mason model, the simulated results indicate a low insertion loss of −1.804 dB at the center frequency of the filter (3.1 GHz), and a good out-of-band rejection of −30 dB. The attenuation values at the left and right sidebands are −1.183 dB (@3.0 GHz) and −1.674 dB (@3.2 GHz), respectively. Additionally, the return loss (*S*_11_) within the passband is consistently below −10 dB.

Next, a combined acoustic–electromagnetic method is adopted to compare the simulation results with those obtained from the circuit model based on pure series-parallel resonators. As shown in Figure 2a, a 3D-electromagnetic simulation model of the filter is established in ANSYS Electronics Desktop software. The series and parallel FBARs are distinguished by different colors. The signal and ground pads are marked with “G“ and “S”, respectively. The “G-S-G” signal ports on the left and right sides of the filter indicate the input and output of the RF signal. Combining the simulation results of the Mason model with those of the electromagnetic model enables the realization of acoustic–electromagnetic co-simulation [6,32,33].

Figure 2b,c demonstrate the filter’s transmission response under the influence of both acoustic and electromagnetic effects. Compared to circuit simulation using the Mason model, the out-of-band rejection of the filter in Figure 2b shows minimal change, with the transmission zeros on both sides shifting upwards. The transmission zero on the left side shifts upward by approximately 65 dB, while that on the right side shifts upward by about 57 dB. This phenomenon arises due to the consideration of parasitic electromagnetic coupling effects present in the filter during electromagnetic simulation. These effects encompass electromagnetic interactions between resonators as well as between the filter and the substrate.

Furthermore, higher electrode loss leads to degradation in the filter’s in-band insertion loss. Due to the consideration of electrode loss across the entire model in electromagnetic simulation, the in-band insertion loss at the filter’s center frequency transitions from −0.867 dB in circuit simulation to −1.551 dB. Additionally, the insertion loss at the passband edge also decreases to −2.453 dB (@3.0 GHz) and −2.588 dB (@3.2 GHz), respectively. The results of the acoustic–electromagnetic co-simulation reveal that, in comparison to circuit simulation using the Mason model, the filter exhibits deteriorated in-band insertion loss and out-of-band rejection performance.

Figure 3 illustrates the simulated temperature distribution on the top of the filter for the power load of 30 dBm. When the power signal is loaded onto the filter from the left input port, the temperature of the first series resonator is significantly higher than that of the other resonators within the filter. As shown in Figure 3, the simulation results indicate that the temperature of the first series resonator even exceeds 50 °C, while the temperatures of the other resonators remain below 30 °C. It is possible that the first series resonator may reach the highest temperature earliest under filter operation.

The detailed microfabrication process flow of FBAR filters is illustrated in Figure 4. Firstly, an 8-inch high-resistivity silicon substrate is etched to form an air cavity. Then, an SiO_2_ film is deposited on the substrate to fill the entire air cavity by plasma-enhanced chemical vapor deposition (PECVD). Next, the SiO_2_ is etched, and excess SiO_2_ is removed using chemical mechanical polishing (CMP). After that, a 139 nm thick Mo is deposited and etched to form the bottom electrode. Subsequently, a 780 nm thick Sc_0.2_Al_0.8_N piezoelectric layer is deposited and etched to expose the pad of bottom electrodes. In the next step, a 43 nm thick Mo is deposited and etched to form a mass loading layer, followed by the deposition and etching of a 135 nm thick Mo to serve as the top electrode. After a 1 μm thick Au film is deposited and patterned to define the probing pad, the release holes are etched and the air cavity is released by vapor hydrofluoric acid (VHF), thus finishing the fabrication of the proposed filter.

## 3. Results and Discussion

As shown in Figure 5a, the X-ray diffraction (XRD) 2*θ*/ω patterns of the Sc_0.2_Al_0.8_N film and Mo film during fabrication were measured to characterize their crystal quality. The rocking curve of the ScAlN (002) peak was also measured as shown in Figure 5b. The full width at half maximum (FWHM) of the (002) peak in the Sc_0.2_Al_0.8_N film is 1.7°. The high-resolution cross-sectional transmission electron microscopy (TEM) image is shown in Figure 5c, where the Sc_0.2_Al_0.8_N film demonstrates favorable crystal orientation along the (002) *c*-axis. The selected area electron diffraction (SAED) result is shown in Figure 5d, with the distance between the two closet diffracted spots and the central spot measuring 7.805 1/nm; this corresponds to a lattice plane (002) distance of 0.256 nm, which is slightly larger than that of AlN (0.249 nm) [34,35,36].

Figure 6a illustrates a cross-sectional view of the fabricated FBAR, with the thickness and materials of each layer already labeled in the figure. It can be seen that the thickness of each layer is approximately close to the designed thickness (see Table 1), indicating delicate control over the thickness of the thin films. The top view of the fabricated FBAR is shown in Figure 6b. It can be clearly observed that the resonator region is connected to the signal side through Mo. The four release holes at the corners of the resonator area are designed to etch the sacrificial layer sufficiently to form a resonant cavity.

The frequency responses of the fabricated FBARs were measured using a Cascade Microtech GSG probe station in connection with a Keysight Network Analyzer (N5222B). The measured impedance curves and conductance curves of the series and parallel resonators are shown in Figure 6c. To achieve better suppression in the filter, the area of the series FBAR is smaller than that of the parallel FBAR, resulting in a larger impedance for the series FBAR [37]. It can be seen that the series FBAR exhibits more spurious modes than the parallel FBAR; this can be attributed to the higher impedance in series FBARs, where spurious modes below *f*_s_ are more easily excited [38].

The corresponding measured parameters of the series and parallel FBARs are summarized in Table 2. The effective electromechanical coupling coefficient ($k_{eff}^{2}$) and quality factor (*Q*) can be calculated by the following formulas [30]:(1)$$k_{eff}^{2}=\frac{\pi^{2}}{4}\cdot \frac{f_{s}}{f_{p}}\cdot \frac{f_{p}-f_{s}}{f_{s}}$$
(2)$$Q\left(f\right)=\frac{2\pi f\tau \left(f\right)\left|S_{11}\right|}{1-{\left|S_{11}\right|}^{2}}$$
where τ(f) is the group delay of *S*_11_.

However, the measured frequencies of both series and parallel FBARs do not perfectly match the simulated results shown in Figure 1b, with slight discrepancies present. This can be attributed to the small difference in material parameters between the simulated model and the actual device. More precise material parameters would contribute to enhancing the accuracy of the simulation results.

Figure 7a demonstrates a top view of the fabricated FBAR filter, where it can be seen that the filter surface appears to be intact, showing no signs of structural damage and exhibiting good manufacturing quality. The measured transmission response is shown in Figure 7b,c. The measured insertion loss is −2.431 dB at 3.090 GHz, and the measured −3 dB bandwidth is 115 MHz (from 3.013 GHz to 3.128 GHz). The rejection values at the edges of the designed frequency band (3.0 GHz–3.2 GHz) are −7.858 dB and −7.594 dB, respectively.

Compared to the results of acoustic–electromagnetic co-simulation, significant discrepancies are observed in the measured results of the fabricated filter, primarily due to two main factors:

On the one hand, there exists a significant disparity between the material parameters employed during the simulation process and those of the materials prepared. Particularly, the material loss incurred during the actual fabrication is notably higher compared to the assumed material losses in the simulation, resulting in a decline in various aspects of the filter’s performance. In general, the damping present in the FBAR significantly affects its *Q*-factor [39,40], which, in turn, greatly influences the in-band characteristics of the filter [30,41,42]. A higher *Q*-factor generally results in better in-band characteristics for the filter [30,41,42]. Upon comparing the measured results of the filter with the simulation results, it appears that this discrepancy is likely due to the damping in the fabricated devices being greater than the damping set in the simulation. This discrepancy is more likely attributable to fabrication issues, which leads to a lower *Q*-factor and, consequently, a larger discrepancy between the measured in-band characteristics of the filter and the simulation results. Furthermore, there are some losses overlooked during the simulation process, such as losses occurring between heterogeneous interfaces within the device and other internal losses within the materials. Multiple processing iterations may be able to reduce these discrepancies.

On the other hand, during the actual manufacturing process, various factors may lead to dimensional discrepancies between the fabricated device and the design model, such as differences in the dimensions of the bottom electrode relative to the cavity edge, and the tilt angle of the bottom electrode. Variations in critical dimensions can affect the reflection of sound waves within the resonator, leading to the emergence of parasitic modes, thereby exerting an influence on the filter’s performance.

Figure 8a shows the power capacity test system of the FBAR filter. The signal generator frequency is adjusted to the frequency point on the left side of the filter’s −3dB passband, and the output power is swept to explore the 1-dB compression point (P1dB) [43]. The power amplifier and power meter are controlled using software to maintain input power levels between 20 dBm and 37 dBm, with a step size of 0.5 dBm. The input power signal is connected to the RF GSG probe through the RF cable via the coupler, thereby applying the signal to the FBAR device. In addition, the output power is detected by the control software. Figure 8b plots the variation of output power with respect to changes in input power; the measured 1-dB compression point (P1dB) is 30.5 dBm.

As the input power continues to increase, the temperature of the device will gradually rise. According to the simulation in Figure 3, it can be inferred that the first series resonator will reach its maximum tolerable temperature first, and a further increase in power will cause it to be burned or damaged. In this work, as shown in Figure 8c, with the power continuing to increase, the first series resonator burns out, which is mutually confirmed with the simulation in Figure 3. Additionally, as illustrated in Figure 8b, there is a sudden decrease in the output power when the input power reaches 31 dBm. Since the input power increment during the testing process is 0.5 dBm, the critical power at which the first series resonator does not burn out should lie between 30.5 dBm and 31 dBm. Therefore, in future filter designs, it is necessary to give priority to the first series resonator. Increasing the maximum power capacity of the first series resonator will help improve the power capacity of the filter. This holds significant importance for the design of high-power filters in both 5G and 6G communication systems.

## 4. Conclusions

In this work, we propose a film bulk acoustic-wave filter for the 3.0 GHz–3.2 GHz S-band. The filter design starts with a circuit design based on the Mason model, employing a four-stage ladder type, with a minimum insertion loss of −0.87 dB. Subsequently, the combined acoustic–electromagnetic method is employed for simulation, with the simulation results being compared to those of the Mason equivalent circuit model. Due to the inclusion of parasitic electromagnetic coupling effects within the filter, the electromagnetic simulation results in the degradation of suppression at the transmission zeros. Furthermore, the presence of electrode loss causes reductions of 0.68 dB, 1.27 dB, and 0.92 dB, respectively, in passband insertion loss at the center frequency and suppression at the upper and lower sidebands. The temperature distribution simulation of the filter reveals that the first series resonator exhibits the highest temperature under filter operation. The proposed FBAR and filter are fabricated using an eight-step lithographic process; the measured results indicate that FBAR has a $k_{eff}^{2}$ of 13.3%, and the minimum passband insertion loss of the filter is −2.4 dB, with a −3 dB bandwidth of 115 MHz (from 3.013 GHz to 3.128 GHz). The measured 1 dB compression point of the fabricated filter is 30.5 dBm. Moreover, as the input power increases further, the first series resonator burns out first, consistent with the temperature distribution simulation results. Therefore, particular attention should be paid to the design of the first resonator in the filter to enhance the filter’s power capacity. This work provides a research foundation for improving the power capacity of RF filters, which can be utilized in the research of filters for 5G and 6G communications.

## Figures and Tables

**Figure 1 sensors-24-02939-f001:**
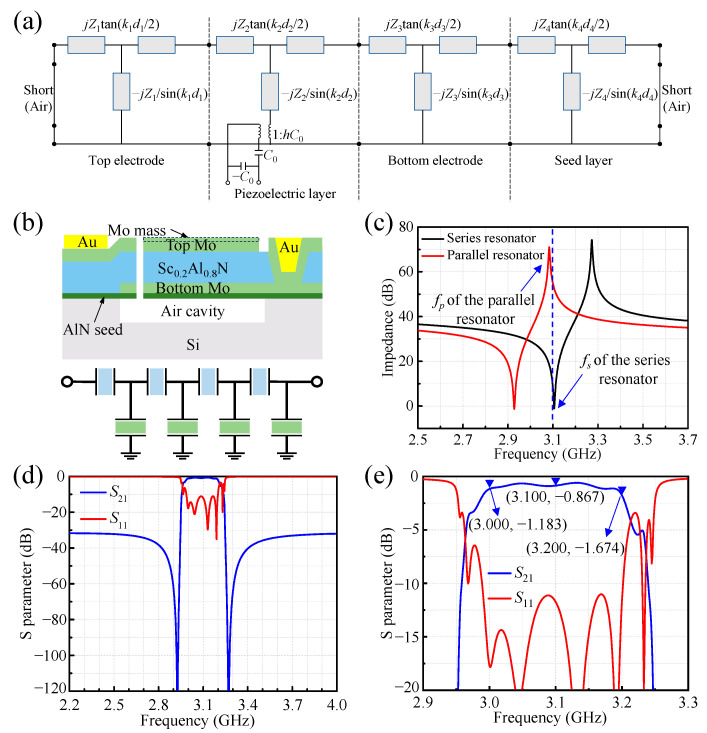
The design of the proposed FBAR and filter: (**a**) Mason model. (**b**) A cross-sectional view of the designed FBAR, and a schematic diagram of the ladder circuit for the filter. (**c**) The simulated frequency responses of FBARs. The simulated electrical response of the filter: (**d**) overall characteristics; (**e**) in-band characteristics.

**Figure 2 sensors-24-02939-f002:**
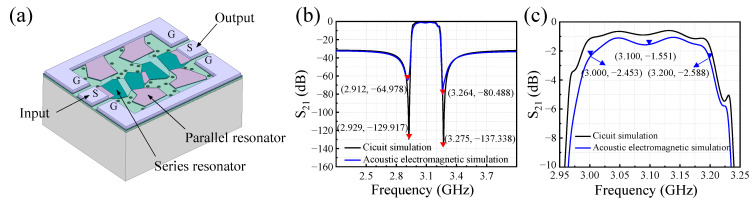
Acoustic–electromagnetic co-simulation of the filter. (**a**) Electromagnetic model of the ladder-type filter based on FBARs. (**b**) Simulated overall transmission response of the filter. (**c**) Comparison of in-band insertion loss between acoustic–electromagnetic co-simulation and circuit simulation results.

**Figure 3 sensors-24-02939-f003:**
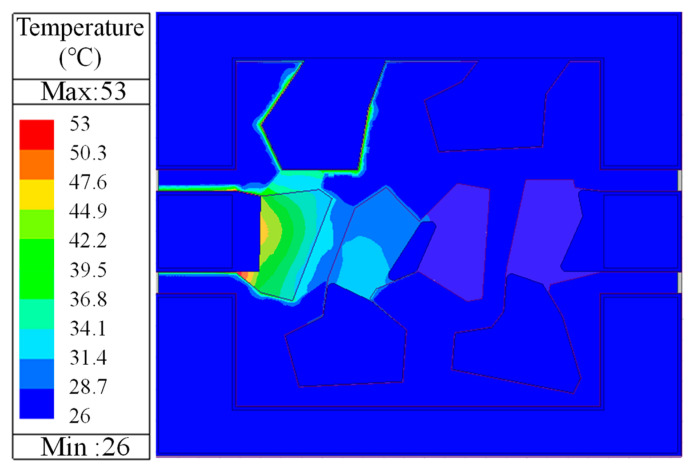
Simulation results for the temperature distribution in the filter for the power load of 30 dBm.

**Figure 4 sensors-24-02939-f004:**
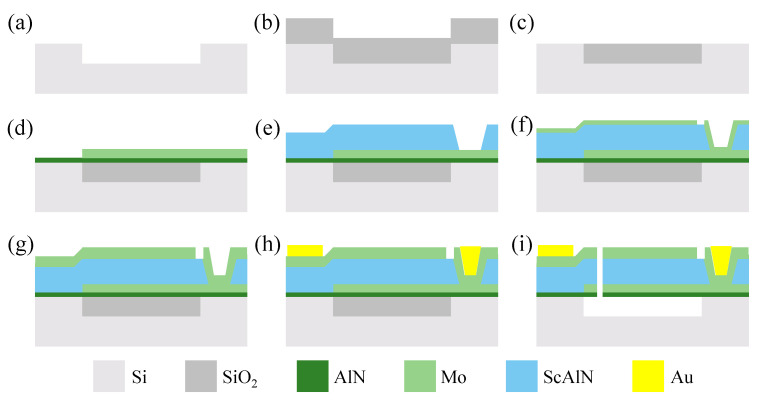
Main process steps for the fabrication of Sc_0.2_Al_0.8_N-based filters. (**a**) Etch Si to form an air cavity. (**b**) Deposit SiO_2_. (**c**) Etch SiO_2_ and CMP. (**d**) Deposit the AlN seed layer and Mo; then, etch Mo to form the bottom electrode. (**e**) Deposit Sc_0.2_Al_0.8_N and etch. (**f**) Deposit Mo; then, etch Mo to form a mass loading layer. (**g**) Deposit Mo; then, etch Mo to form the top electrode. (**h**) Deposit Au; then, pattern it to form the electrode pad (lift-off). (**i**) Etch release holes; then, release the SiO_2_ to form an air cavity.

**Figure 5 sensors-24-02939-f005:**
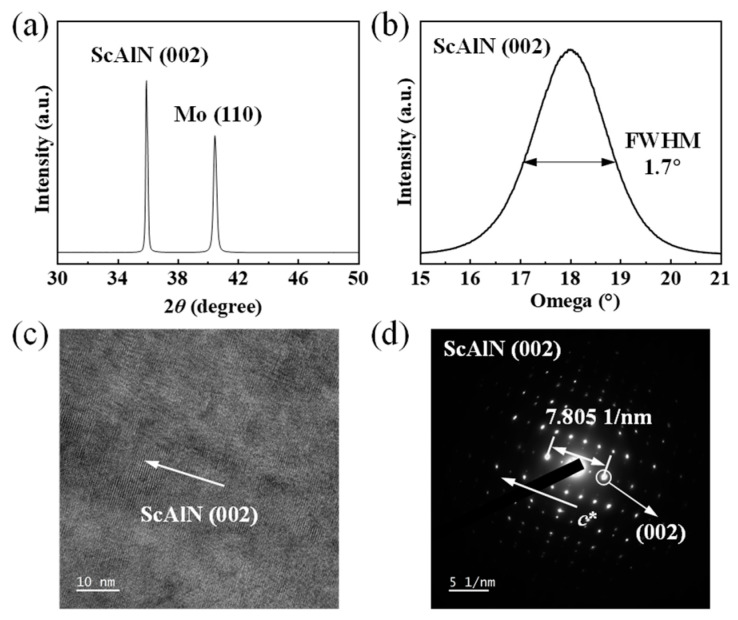
(**a**) XRD patterns of Sc_0.2_Al_0.8_N film and Mo film during fabrication. (**b**) Rocking curve of ScAlN (002) peak. (**c**) SAED pattern of Sc_0.2_Al_0.8_N film. (**d**) High-resolution TEM image of Sc_0.2_Al_0.8_N film.

**Figure 6 sensors-24-02939-f006:**
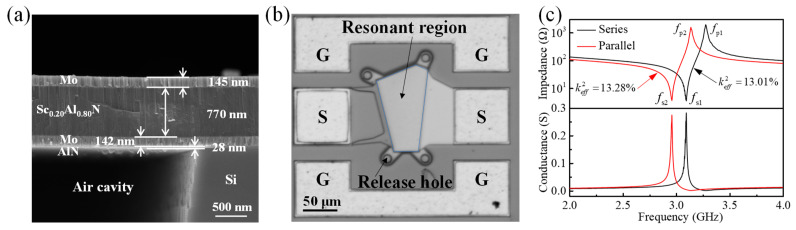
(**a**) Scanning electron microscope (SEM) image of the cross-sectional view of the fabricated FBAR. (**b**) Top view of the fabricated FBAR. (**c**) Measured impedance curves and conductance curves of the series and parallel FBARs.

**Figure 7 sensors-24-02939-f007:**
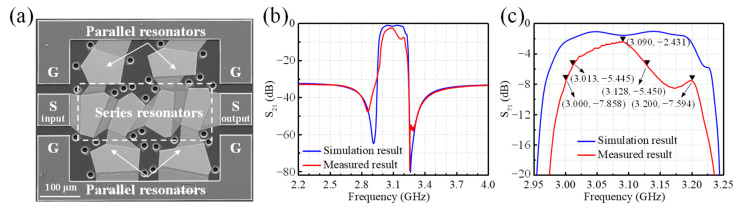
(**a**) SEM image of the fabricated FBAR filter. Comparison of S_21_ parameter between acoustic–electromagnetic co-simulation result and measured result of the filter transmission response: (**b**) overall characteristic, (**c**) in-band characteristic.

**Figure 8 sensors-24-02939-f008:**
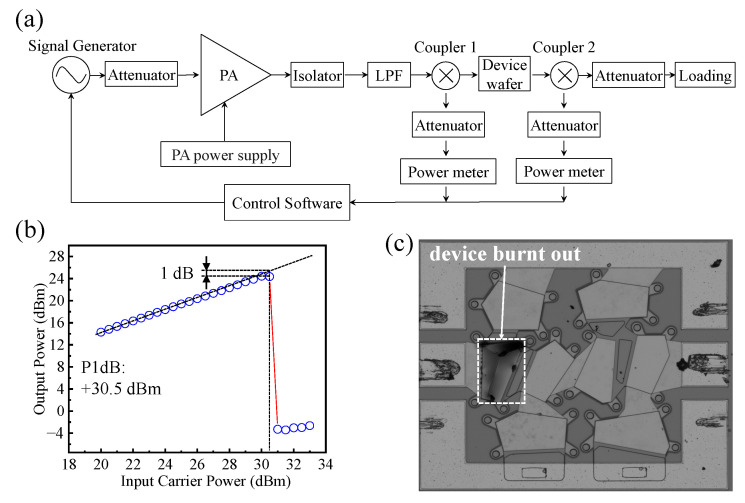
(**a**) Power capacity test system. (**b**) The output power against the input power. (**c**) Device burnout caused by excessive power.

**Table 1 sensors-24-02939-t001:** The materials and corresponding thicknesses used in the filter design.

Resonator	AlN Seed Layer	Bottom Mo	Sc_0.2_Al_0.8_N	Top Mo	Mo Mass Loading
Series	25 nm	139 nm	780 nm	135 nm	0
Parallel	25 nm	139 nm	780 nm	135 nm	43 nm

**Table 2 sensors-24-02939-t002:** The measured parameters of the fabricated series and parallel FBARs.

Resonator	*f_s_* (GHz)	*f_p_* (GHz)	$k_{eff}^{2}$ (%)	*Q* _s_	*Q* _p_
Series	3.0936	3.2760	13.0	295	209
Parallel	2.9586	3.1372	13.3	267	217

## Data Availability

The data are contained within the article.

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
