# Peer review of "Design and Fabrication of a Film Bulk Acoustic Wave Filter for 3.0 GHz–3.2 GHz S-Band"

_sensors, 2024, doi:10.3390/s24092939_

Round 1

Reviewer 1 Report

Comments and Suggestions for Authors

The paper is very well presented. The technical explanations are very clear. Special care has been taken with the figures, which are of high quality and help to understand what has been done.

The results are interesting.

The weakness of the paper lies in its positioning relative to the state of the art. Many works have been carried out, over the last ten years at least, in the field of AlScN-based F-BAR (check, especially, IEEE Ultrasonics literature). It is important to improve the paper's introduction by citing more works, especially those carried out in Europe and the USA, in this field. It is also important to mention competing approaches, sometimes more sophisticated.

Therefore, please improve the introduction by citing some additional references, to better position the paper with respect to the state of the art in the field of AlScN-based FBARs. What is the contribution of the paper compared to the state of the art? Higher power, in the 3-3.2GHz band? Please also mention some possible applications of your work (e.g., 6G...)

Lines 37-38: Hasty conclusion. To be tempered.

The simulation models used are very well known and have been mastered for a long time. The most interesting results of the paper are those presented in Figure 7. Please superpose the experimental results of Figure 7b on the simulation results to allow for better comparison.

You mention as a possible source of the observed differences, a problem with the calculation of the AlScN coefficients. There is also abundant literature on this subject. See for example: https://doi.org/10.1109/ULTSYM.2018.8579706. Please elaborate on this point, and cite appropriate references.

Comments on the Quality of English Language

The English is very correct. The paper is easy to read. A few sentences could be improved.

Reviewer 2 Report

Comments and Suggestions for Authors

This manuscript presents a detailed analysis of FBAR through simulation and experimental approaches. It is well-organized and well-written. This work provides valuable insights for the FBAR designs. I recommend its acceptance after minor revision. My detailed comments are as follows:

1. Please give a schematic diagram of the Mason model.

2. Please give us the source of the material constants used for the simulation?

3. What are the advantages of this article compared to other similar works.

4. What is the critical power at which the first series resonator is not burned out?

5. In Fig.6(c), the series FBAR seems have more spurious mode. Please show the conductance curve and explain it.
